# Differential Regulation of MMPs, Apoptosis and Cell Proliferation by the Cannabinoid Receptors CB1 and CB2 in Vascular Smooth Muscle Cells and Cardiac Myocytes

**DOI:** 10.3390/biomedicines10123271

**Published:** 2022-12-16

**Authors:** Bettina Greiner, Manuela Sommerfeld, Ulrich Kintscher, Thomas Unger, Kai Kappert, Elena Kaschina

**Affiliations:** 1Cardiovascular–Metabolic–Renal (CMR)-Research Center, Institute of Pharmacology, Corporate Member of Freie Universität Berlin, Humboldt-Universität zu Berlin, Charité—Universitätsmedizin Berlin, 10115 Berlin, Germany; 2DZHK (German Centre for Cardiovascular Research), partner site Berlin, 10115 Berlin, Germany; 3CARIM School for Cardiovascular Diseases, Maastricht University, 6211 LK Maastricht, The Netherlands; 4Clinical Chemistry and Pathobiochemistry, Institute of Diagnostic Laboratory Medicine, Corporate Member of Freie Universität Berlin, Humboldt-Universität zu Berlin, Charité—Universitätsmedizin Berlin, 10117 Berlin, Germany

**Keywords:** cannabinoid receptors, MMP-2, MMP-9, VSMC, H9c2 cells, glucose, cell proliferation, apoptosis, cardiovascular disease

## Abstract

Cannabinoids (CB) are implicated in cardiovascular diseases via the two main receptor subtypes CB_1_R and CB_2_R. This study investigated whether cannabinoids regulate the activity of matrix metalloproteases (MMP-2, MMP-9) in vascular smooth muscle cells (VSMCs) and in cells of cardiac origin (H9c2 cell line). The influence of CB_1_- and CB_2_ receptor stimulation or inhibition on cell proliferation, apoptosis and glucose uptake was also evaluated. We used four compounds that activate or block CB receptors: arachidonyl-2-chloroethylamide (ACEA)—CB_1_R agonist, rimonabant—CB_1_R antagonist, John W. Huffman (JWH133)—CB_2_R agonist and CB_2_R antagonist—6-Iodopravadoline (AM630). Treatment of cells with the CB_2_R agonist JWH133 decreased cytokine activated secretion of proMMP-2, MMP-2 and MMP-9, reduced Fas ligand and caspase-3-mediated apoptosis, normalized the expression of TGF-beta1 and prevented cytokine-induced increase in glucose uptake into the cell. CB_1_R inhibition with rimonabant showed similar protective properties as the CB_2_R agonist JWH133, but to a lesser extent. In conclusion, CB_1_R and CB_2_R exert opposite effects on cell glucose uptake, proteolysis and apoptosis in both VSMCs and H9c2 cells. The CB_2_R agonist JWH133 demonstrated the highest protective properties. These findings may pave the way to a new treatment of cardiovascular diseases, especially those associated with extracellular matrix degradation.

## 1. Introduction

Recent studies have demonstrated hemodynamic and cardiometabolic effects of cannabinoids [1,2,3,4,5]. The cannabinoid system modulates the extracellular matrix turnover in the heart and blood vessels [6,7]. These findings have led to interest in the biochemical bases of their action. Endocannabinoids exert their effects, at least in part, by stimulating two main receptor subtypes, CB_1_ and CB_2_ (CB_1_R and CB_2_R), which belong to a group of seven transmembrane-spanning receptors. They are coupled to Gi/o-proteins and act via an inhibition of adenylylcyclase and subsequently the reduction of cAMP [8,9]. Other than the adenylylcyclase/cAMP pathways, several other intracellular pathways are influenced, such as p38, JNK and ERK [10]. Cannabinoid receptor ligands are divided into endogenous cannabinoids, such as 2-AG and AEA, and exogenous cannabinoids, such as derivatives from the cannabis sativa plant or synthetic cannabinoids [10]. Currently, novel modulators of the cannabinoid system are under investigations. New compounds are able to bind to CB receptors in the low nanomolar range with a marked selectivity towards the receptors [11]. Moreover, multitargeting G-protein-coupled receptors is also a promising strategy, as shown for antinociception by bivalent agonists for the opioid and cannabinoid receptors [12,13]. CB_1_R stimulation elicits bradycardia, negative inotropy and hypotension [14]. The CB_1_R is also implicated in inflammation, apoptosis and oxidative stress in the heart [15], whereas the CB_2_R may play a protective anti-inflammatory, antioxidative and antiatherogenic role [15,16,17]. We have previously shown that blockade of the CB_1_R with rimonabant decreased collagen accumulation and prevented upregulation of the profibrotic protein TGF-β1 in the heart and aorta in a myocardial infarction model [7]. Moreover, CB_1_R blockade also reduced the activity of the matrix metalloprotease 9 (MMP-9) in cardiac fibroblasts [7]. On the other hand, genetic deletion of the CB_2_R increased TGF-β1 and collagen production in a heart failure model [18], pointing to opposite effects of CB_1_R and CB_2_R. Recent studies provided further findings on extracellular matrix regulation by cannabinoids in the heart and vessels. For example, CB_2_R knockout models showed an increase in atherosclerotic vascular changes as well as an increase in MMP-9 expression [19]. In patients with high plaque instability of the carotid artery, decreased CB_2_R expression was correlated with a MMP-9 increase [20]. Based on these findings, we hypothesized that cannabinoids may influence proteolytic processes in the vessels directly *via* CB_1_R and CB_2_R, which are known to be localized on vascular smooth muscle cells (VSMCs) [21,22]. Therefore, we aimed to investigate whether cannabinoids regulate the activity of MMPs in the cells of rat vascular (VSMCs) and cardiac origin (H9c2). Given that the gelatinase A (MMP2) and gelatinase B (MMP9) are capable of degrading components of the extracellular matrix [23], we have focused on their regulation. By using various cannabinoid receptor ligands, we also intended to explore which receptor subtype is implicated in proteolysis. In our study, we used four different synthetic compounds that activate or block cannabinoid receptors: arachidonyl-2-chloroethylamide (ACEA), a CB_1_R agonist; rimonabant, a CB_1_R antagonist; John W. Huffman (JWH133), a CB_2_R agonist; and 6-Iodopravadoline (AM630), a CB_2_R antagonist. Since MMPs secretion is closely connected to cell proliferation, apoptosis and glucose metabolism, we analyzed their regulation.

## 2. Materials and Methods

### 2.1. Cell Cultures

Primary vascular smooth muscle cells (VSMCs) were isolated from the aorta of male normotensive Wistar rats (200 to 220 g; Charles River Laboratories Germany GmbH) as previously described [24]. Briefly, under dissecting microscope fat, connective tissue and outgoing arteries were removed from the aorta. The vessel was cut longitudinally, and the endothelium was removed with a cell scraper by gentle scraping along the luminal face. VSMCs were isolated by using digestion method, cultured and frozen in liquid nitrogen.

The cells were identified by “hill and valley” growth pattern and immunofluorescence staining with an anti-smooth muscle actin monoclonal antibody (Merk KGaA, Darmstadt, Germany). VSMCs were cultured in Dulbecco’s modified Eagle’s medium (4.5 g/L glucose) supplemented with 10% fetal bovine serum. Experiments were performed with cultures from passages 4 to 12.

This study was carried out in strict accordance with national and European guidelines for animal experiments with approval by the ethics commission of the regulatory authorities of the City of Berlin, Germany, the “Landesamt for Gesundheit und Soziales” (registration number G0002/16).

Neonatal rat cardiomyocytes (H9c2 cell line, 88092904, Sigmaaldrich, Merk, Germany) are a subclone of the original clonal cell line derived from embryonic BD1X rat heart tissue that exhibit many of the properties of skeletal muscle. The cells were incubated in high-glucose Dulbecco’s modified Eagle’s medium with 10% fetal bovine serum. Cell splitting was performed when H9c2 cells reached confluence.

### 2.2. Cell Culture Experiments

Confluent cells were serum-deprived for 24 h. The cells were exposed to recombinant interleukin-1α (IL-1α) 1.0 ng/mL (Sigma-Aldrich, Taufkirchen, Germany) to induce secretion of MMPs. Incubation was performed with or without added compounds for 48 h. The following CB_1_R and CB_2_R agonists and antagonists were used: CB_1_R agonist arachidonyl-2-chloroethylamide (ACEA) (0.5 µM), CB_1_R antagonist rimonabant (1.0 µM), CB_2_R agonist JWH133 (0.5 µM) and CB_2_R antagonist 6-Iodopravadoline (AM630) (1.0 µM). Epigallocatechin gallate (ECEG) (5 µg/mL), an inhibitor of MMPs, was used as a positive control. Conditioned media were obtained by collecting the culture media at the end of the experiment. Proteins were extracted from the cells and processed for Western blot analysis.

### 2.3. Chemical Compounds

Arachidonyl-2-chloroethylamide (ACEA), JWH133, and 6-Iodopravadoline (AM630) were purchased from Tocris (Bristol, UK). Rimonabant was purchased from Sanofi Aventis Deutschland GmbH (Frankfurt, Germany) and epigallocatechin gallate (ECEG) from Enzo Life Science (Lörrach, Germany).

### 2.4. Gelatin Zymography

Cultured media harvested from cells were analyzed for MMP2 and MMP9 by gelatin zymography, as described previously [25]. Briefly, conditioned media aliquots were resuspended in nonreducing sample buffer and applied to 10% SDS-PAGE copolymerized with gelatin (1 mg/mL). After electrophoresis, the gels were washed for 1 h in Triton-X-100 (2.5% v/m), incubated overnight in an enzyme buffer (developing buffer) at 37 °C, stained in Coomassie solution for 1.5 h and subsequently destained for 1 h in a destaining solution. By that, the enzymatic active areas became visible as a transparent band on the blue-stained gel. The zymograms were analyzed with Scion ImageJ software.

### 2.5. Western Blot Analysis

Protein samples were separated via SDS-PAGE and transferred to Amersham Hybond PVDF membranes (VWR International, LLC, Radnor, PENN, USA). Membranes were probed with antibodies against MMP-9, caspase-3 (1:1000 in 1× TBST with 5% *w*/*v* nonfat dry milk) (Abcam, Hiddenhausen, Germany), FasL, TGF-beta1 (1:500 in 1× TBST with 5% *w*/*v* nonfat dry milk) (Santa Cruz Biotechnology Inc, Heidelberg, Germany) and then incubated with peroxidase-conjugated secondary antibodies anti-mouse antibody/anti-goat antibody (1:2000 in 1× TBST with 5% *w*/*v* nonfat dry milk) (Agilent Dako, Santa Clara, CA, USA). Protein expression was normalized to glyceraldehyde-3-phosphate dehydrogenase (GAPDH) (Abcam, Hiddenhausen, Germany). Immunoreactive bands were detected by enhanced chemiluminescence (GE Health Care, Solingen, Germany) and quantified with ImageJ Fiji software.

### 2.6. Immunofluorescence

Protein expression of FasL, caspase-3 and TGF-beta1 were studied by fluorescence microscopy (Biorevo BZ-9000, Keyence, Japan) on cover slips with cells after stimulation with IL-1α (1.0 ng/mL) in the presence or absence of compounds after 48 h. After treatment, cell permeability was increased by Triton-x (5%). Afterwards the primary antibody was added (1:100 in 5% donkey serum), caspase-3, FasL and TGF-beta1 overnight. After washing, the secondary antibody was added (1:100; FITC/Cy3, Agilent Dako, Santa Clara, CA, USA). Nuclei were stained with Hoechst (1:4000) (Merck Sigma-Aldrich, Darmstadt, Germany).

### 2.7. IncuCyte Live-Cell Analysis

IncuCyte Live-Cell Analysis System (IncuCyte ^®^ S3, Sartorius, Göttingen, Germany) was used to observe cell death and proliferation. The experiments were performed with VSMCs and H9c2 cells. Cells were transferred into 96-well plates and treated with the compounds or vehicle under same conditions (medium with 1% or 10% FBS) for 48 h (up to 5 days). Cell proliferation was monitored by analyzing the occupied area (% confluence) of cell images over time. Analysis of the IncuCyte images was performed with Incucyte^®^ Analysis Software.

### 2.8. Glucose, Lactate, Electrolytes Concentrations

The concentrations of glucose, lactate, sodium, calcium and potassium were measured in cell supernatant 48 h after treatments. Na, K and Cl were quantified by an indirect ion selective electrode (ISE, Gen. 2 Roche^®^ Diagnostics GmbH) on a Roche^®^ cobas ISE module. Lactate dehydrogenase activity was determined applying Roche^®^ cobas lactate dehydrogenase according to IFCC version 2 (LDHI2, #05169330 190). Glucose and lactate were quantified photometrically on a Roche^®^ cobas analyzer.

### 2.9. Statistical Analysis

Results are expressed as the mean ± (SD) standard deviation in the graphics unless declared otherwise in the figure legends. Two-group comparisons were analyzed by the 2-tailed Student unpaired *t*-test for independent samples. Welch’s correction was used when results in relation to IL-1α were compared, as declared in the figure legends. Pearson R statistical test was used for measuring the strength between glucose and lactate variables and their relationship. Statistical analysis was performed using GraphPad Prism 6 (GraphPad Software Inc., La Jolla, CA, USA).

## 3. Results

### 3.1. Regulation of MMP-2 and MMP-9

MMPs’ enzymatic activity levels in conditioned media were demonstrated by gelatin zymography. Experiments were performed in different passages of VSMC (P4-P12, *n* = 10–16). The results are presented in Figure 1.

MMP-9 showed a greater upregulation by IL-1α stimulation (Figure 1a; 92.1%, *p* < 0.0001) than proMMP-2 (Figure 1b; 39.7%, *p* < 0.001) and MMP-2 (Figure 1c; 13.6%, n.s.). In comparison to the IL-1α group, treatment with JWH-133 reduced MMP-9 activity by 30.4% (Figure 1a; *p* < 0.0001). Rimonabant, ACEA and AM630 reduced MMP-9 activity slightly but not significantly. Even though less upregulated by IL-1α, proMMP-2 was also reduced by JWH-133 treatment (Figure 1b; 27.6%; *p* < 0.05). Rimonabant reduced proMMP-2 by 14.2%, while AM 630 and ACEA decreased proMMP-2 only by 7.7% and 7.9%, respectively. These changes were statistically not significant.

MMP-2, while showing only a minor upregulation after IL-1α stimulation, was also reduced by JWH-133 treatment (Figure 1c; 13.2%; *p* < 0,05). Rimonabant reduced MMP-2 by 10.1%, AM 630 by 11.2% and ACEA only by 6.3%, showing no significant effects. Representative zymographies are shown in Figure 1d,e.

The same experimental setup was used to evaluate MMP activity in H9c2 cardiac cells after treatment with the cannabinoid receptor agonists and antagonists after IL-1α stimulation. Similar to VSMC, MMP activity was more affected by the treatment with the CB_2_R agonist JWH-133, which decreased MMP-9 by 12.6% and proMMP-2 by 29.9% (n.s., Appendix A).

MMP-2,9 protein expression analysis is presented in Figure 2. Apart from the proMMP-9 (92 kDa) and MMP-9 (72 kDa) bands, three bands in the area of 45 to 60 kDa were detected (Figure 2a). ProMMP-9 expression increased by 48.4% after stimulation with IL-1α compared to control (Figure 2b), and the CB_2_R agonist JWH-133 augmented the increase by 35.5%, whereas the CB_2_R antagonist AM 630 increased the expression of proMMP-9 by 50.8%. Moreover, proMMP-9 expression was increased by rimonabant (+4.5%) and CB_1_R agonist ACEA (+13.0%). The bands in the area of 45–60 kDa, which were not expressed in the control group, showed a higher degree of regulation than the proMMP-9 and MMP-9 bands (Figure 2b). The CB_2_R agonist JWH-133 reduced the MMP expression of the three bands by 60.7%, and the CB_1_R antagonist rimonabant reduced MMP expression by 25.7%. The correlating agonists/antagonists increased MMP activity at this molecular weight, AM630 by (+46.2%) and ACEA by (+40.9%).

### 3.2. Regulation of Apoptosis

#### 3.2.1. Apoptosis Ratio

MMP secretion is closely connected with cell death mechanisms. Therefore, the experimental setup used for zymography was also used to perform cell nuclei staining to evaluate apoptosis. The cells were treated with different compounds and cultivated on cover slips. After stimulation of VSMC with IL-1α for 48 h, apoptosis was increased, as demonstrated by an apoptotic ratio (Figure 3). More condensed small cell nuclei with irregular form (blebbing) as well as more disintegrating cells were detected (Figure 3A). IL-1α stimulation increased the number of apoptotic cell nuclei as compared with control. JWH-133 and rimonabant partially mitigated this effect. Rimonabant increased the ratio of normal cell nuclei to apoptotic cell nuclei by 2.5-fold and JWH-133 by 2.0-fold compared to IL-1α-treated cells (Figure 3B). Treatment with ACEA and AM630 did not have a similar effect; apoptosis levels in those two groups were comparable to that after IL-1α stimulation (Figure 3B; IL-1α = 1.0).

The same experimental setup was performed in H9c2 cells. The proapoptotic effect of the IL-1α stimulation was less pronounced in comparison to the IL-1α stimulation in VSMC, hence the effect of the treatment also showed smaller effects. IL-1α treatment increased the number of apoptotic cells 1.2-fold as compared to the control. The CB_2_R agonist JWH-133 ameliorated this increase 1.1-fold and rimonabant 1.3-fold, while AM 630 treatment showed no difference to the IL-1α group and ACEA showed even more apoptotic cell nuclei than the IL-1α-stimulated H9c2 cells (Appendix A).

#### 3.2.2. Regulation of Caspase-3, FasL and TGF-Beta1

We further investigated the expression of apoptotic markers caspase-3 and FasL as well TGF-beta1 by fluorescence staining in VSMC. The intensity of caspase-3 expression in different groups is shown in Figure 4A. caspase-3 staining was localized intracellularly in VSMCs. IL-1α showed similar levels of caspase-3 signal compared with the control group. Rimonabant increased caspase-3 expression (2.3-fold), and the CB_1_R antagonist ACEA showed an even higher caspase-3 signal (2.8-fold) as compared with IL-1α group. The CB_2_R antagonist AM 630 showed an increased caspase-3 signal (1.5-fold), whereas JWH-133 decreased caspase-3 expression under the IL-1α level (by 43.2%) as well as under the control level. Thus, it appears that CB_2_R activation reduced apoptosis via caspase-3 signaling.

FasL staining was strongly increased by 75.5% after IL-1α stimulation compared with the control (Figure 4B). CB_1_R blockage with rimonabant showed a decrease of 57.0% in FasL signal compared to IL-1α, while IL-1α plus ACEA showed the highest expression of all treatment compounds (Figure 4B(a,b)). Treatment with JWH-133 decreased FasL fluorescence signal, 4.5-fold compared to IL-1α, even under control group levels (Figure 4B(a,b)). This result suggests an antiapoptotic effect mitigated via CB_2_R activation.

Interestingly, TGF-beta1-expression staining showed opposing results compared to FasL expression (Figure 4C). The involvement of TGF-beta1 in the regulation of cell apoptosis has long been a point of discussion, since it contributes to a plethora of processes in the cell. Our results showed a 3.4-fold downregulation of TGF-beta1 in the IL-1α group compared to control (*p* < 0.001) (Figure 4C). Rimonabant and JWH-133 (*p* < 0.005) reduced this decrease (Figure 4C), while treatment with AM 630 and ACEA even decreased TGF-beta1 as compared to the IL-1α-stimulated group (19.1%, and 11.0%, respectively, *p* < 0.001).

### 3.3. Regulation of Cell Proliferation: IncuCyte Live-Cell Analysis

IncuCyte live-cell analysis enables the visualization and quantification of cell behavior over time, thus providing insight into cell proliferation and cell death dynamics.

We performed live-cell analysis using a DMEM cell medium containing either 1% FBS or 10% FBS. In line with our previous experiments for zymography analysis, 1% FBS cell medium was used initially. However, such experimental conditions provoked stagnating confluence due to cell death in all groups including the control. Growth rates without IL-1α stimulation in the first 24 h and 48 h were extremely low in all treatment groups (0.1–5.6%). A minimal cell growth rate was observed in the ACEA group (+0% in 24 h, + 3% after 48 h), being in line with the results obtained in the apoptosis analysis.

In order to improve growth conditions, repeated experiments were performed using 10% FBS medium (Figure 5A). A 10% FBS medium increased the growth rates of the control cells in comparison with 1% FBS (14.1% vs. 5.6% after 24 h and 28.0% vs. 5.1% after 48 h). The growth rate of the VSMC after 24 h and 48 h after using 10% FBS DMEM are presented in Figure 5A(a). IL-1α increased the growth rate in comparison to the control in 24 h (*p* < 0.001). Growth rates equalized after 48 h and showed no significant differences between cannabinoid treatment groups without IL-1α stimulation and the control group (Appendix A).

We repeated the experiment using IL-1α stimulation in cannabinoid treatment groups. IL-1α increased cell growth rate after 24 h compared to the control group (Figure 5B(b)), and JWH-133 mitigated this increase (Figure 5B(b); *p* < 0.05). After 48 h, only the control compound ECEG decreased the cell growth (Figure 5B(b); *p* < 0.05) as compared with IL-1α.

Repeating the experimental setup using the cardiac H9c2 cell line, we achieved differing results. H9c2 as a secondary cell line showed an increased growth rate. After treatment, similar growth rates leading to full confluence in all groups were observed with and without IL-1α stimulation (Appendix A).

### 3.4. Regulation of Glucose, Lactate and Electrolytes

In order to ascertain if the treatment compounds influenced cell metabolics, we measured glucose, lactate and electrolytes concentrations in the supernatant. IL-1α stimulation decreased glucose concentration in the supernatant of VSMCs (Figure 6a; *p* < 0.01). JWH-133 and rimonabant normalized glucose levels, as compared to IL-1α stimulation, up to the control levels (Figure 6a; *p* < 0.05). ACEA and AM 630, in contrast, showed no significant effects (Figure 6a).

Concomitantly, lactate concentration was increased after the IL-1α stimulation (Figure 6b; 2.2-fold). Rimonabant (by 46.5%) and JWH-133 (by 52.7%) reduced this increase (Figure 6b). The correlation analysis confirmed a negative correlation between glucose levels and lactate levels in the JWH-133 group (r = −0.99; *p* < 0.05) and AM630 group (r = −0.99; *p* < 0.01).

Similar measurements were performed in the supernatant of treated H9c2 cells (Figure 6c,d). JWH-133 also reduced the decrease in glucose concentration after IL-1α stimulation (Figure 6c; *p* < 0.05). In H9c2 cells, lactate levels showed less scattering and similar levels in all treatment groups (Figure 6d). The electrolytes Na, K and Cl were not affected by the different treatments, neither in VSMCs (Figure 7a–c) nor in H9c2 (Appendix A).

A comparison of the CB1 and CB2 receptor agonists and antagonists on cytokine-induced MMPs secretion, apoptosis, glucose uptake and cell proliferation in VSMCs and cardiac H9c2 cells is presented in Table 1.

## 4. Discussion

CB receptors are implicated in cardiovascular patho/physiological processes [1,2,3,5], in particular, in the degradation of the extracellular matrix (ECM) [6,7]. The vascular and cardiac cells could be regulated by these receptors, affecting cell metabolism, proteolytic processes, cell death and proliferation.

In the present study, by using various CB receptor ligands, we intended to find out which receptor subtype is implicated in proteolysis in VSMC. We demonstrated that both CB receptors are involved in the regulation of MMPs, although the CB_2_R subtype plays a more important role. In our study, the stimulation of the CB_2_R in the VSMCs by the agonist JWH-133 reduced MMP-9 secretion in the supernatant and decreased proMMP-9 protein expression in the cells. In contrast, the CB_2_R antagonist AM630 increased MMP-9 expression.

Gelatinase MMP-2 is known to be involved in the degradation of extracellular matrix components and angiogenesis [23]. JWH-133 induced a reduction of both proMMP-2 and MMP-2. The same tendency was demonstrated in H9c2 cells of cardiac origin. Thus, we provide evidence that CB_2_R stimulation prevents the cytokine-induced MMP-9 and MMP-2 secretion. Our finding is in line with a study from [20] in neutrophils, which showed that treatment with JWH-133 reduced the release of TNF-α-induced MMP-9 via ERK1/2 phosphorylation. Moreover, JWH-133 exhibited antiproteolytic effects against MMP-1 and MMP-3 in human tenon fibroblasts [26].

CB_1_R inhibition with rimonabant in the present study also tended to decrease MMP-9 secretion, confirming our previous data obtained in cardiac fibroblasts [7]. Despite this, an opposite regulation by the CB_1_R agonist ACEA could not be shown.

In summary, CB_2_R stimulation decreased proteolytic activity in VSMC, mainly by downregulation of MMP-9.

Given the multiple roles of MMPs in cell death and especially in apoptosis, we decided to gain further insights in the effects of the CB agonists and antagonists on apoptosis. The CB_2_R agonist JWH-133 as well as the CB_1_R antagonist rimonabant mitigated the cell-damaging apoptotic effect of cytokine stimulation in VSMC, as demonstrated by nucleus staining. Further, we could show that the CB_2_R agonist JWH-133 reduced the expression of apoptotic markers caspase-3 and FasL, whereas the CB_2_R antagonist showed increased caspase-3 expression.

Our results concerning the role of the CB_2_R in apoptosis are in harmony with studies on cell survival performed in cardiac myocytes and fibroblasts [15] and in the heart ischemia–reperfusion model [27]. Moreover, our data on the CB_1_R are also in line with the findings that deletion of CB_1_R or treatment of diabetic mice with CB_1_R antagonist SR141716 prevented retinal cell death [28].

Nevertheless, the experiments in the H9c2 cell line failed to demonstrate a regulation of apoptosis via CB receptors. Given that these cells showed a decreased reactivity to the IL-1α stimulation, another protocol of apoptosis induction should be tested in future investigations.

TGF-beta1 is an important multifunctional cytokine, which is implied in extracellular matrix remodeling, cell proliferation and cell apoptosis [29]. Interestingly, TGF-beta1 expression was decreased after IL-1α stimulation, in contrast to MMP-9, MMP-2 regulation and the apoptosis rate. These findings are in agreement with Risinger G.M. et al. [30], who showed that TGF-beta1 suppresses the upregulation of MMP-2 by VSMCs. Notably, the CB_1_R antagonist rimonabant and the CB_2_R agonist JWH-133 normalized TGF-beta1 expression in the VSMCs up to the control levels.

VSMC proliferation is known to be important for vascular wall remodeling in response to injury. Given that the amount of secreted MMPs depends on cell number, the effect of the treatment protocols on cell proliferation was studied. Therefore, IncuCyte cell life analysis was used to obtain data on the dynamics of cell proliferation and cell death. Neither the CB_2_R agonist JWH-133 nor the CB_1_R antagonist rimonabant significantly influenced cell proliferation under the given experimental conditions, suggesting that MMP secretion and apoptosis are regulated by the CB receptors. Nevertheless, at higher FBS concentrations (10%), JWH-133 at one time point 24 h showed antiproliferative properties. Our results may partly explain the controversy from previous studies showing pro-proliferative [22] and antiproliferative [16,31] effects on CB_2_R stimulation. Interestingly, we also found a strong antiproliferative effect of ECEG that has been used as a control substance in our experimental setting. Thus, further investigations on the role of the CB_2_R as well as ECEG in the atherosclerosis, angiogenesis and tumor growth would be important.

Since cell metabolism is an essential link between apoptosis and cell proliferation [32], we also addressed the regulation of glucose, lactate and electrolytes after treatment. The electrolytes sodium, potassium and chloride were not affected by treatment with CB_1_R and CB_2_R agonists and antagonists. IL-1α stimulation strongly decreased glucose concentration in the supernatant in comparison to the control group. Such decrease can be explained by an increase in glucose uptake into the cell due to activation of glucose transporters GLUT1/4, which are predominant transporters in VSMCs [33,34]. CB_2_R stimulation with JWH-133 as well as CB_1_R inhibition with rimonabant reduced the decrease in glucose levels of the supernatant, pointing to a possible interaction of the CB receptors with glucose transporters. Concomitantly, the concentration of lactate was increased after IL-1α stimulation, and rimonabant and JWH-133 also reduced this increase. The effects of JWH-133 on glucose levels were confirmed in cardiac H9c2 cells.

Whether glucose regulation by CB receptors is primary to MMPs secretion requires further investigation. Metabolic changes in VSMC not only contribute to the regulation of cell proliferation, apoptosis and proteolysis but also regulate a switch from the “contractile” phenotype to the proliferative “synthetic” VSMC phenotype [35], thereby influencing the progression of vascular diseases. Therefore, the involvement of the CB receptors in the regulation of glucose metabolism is of relevance in the context of several vascular diseases, including atherosclerosis, diabetes, hypertension and aneurysms.

In summary, the CB_1_R and the CB_2_R exert opposite effects on the regulation of cell glucose metabolism, proteolysis and apoptosis in VSMCs and cardiac H9c2 cells.

The stimulation of the CB_2_R reduced the cytokine-activated secretion of proMMP-2, MMP-2 and MMP-9, reduced FasL and caspase-3 mediated apoptosis, normalized the expression of TGF-beta 1 and prevented cytokine-induced increase in glucose uptake into the cell. CB_1_R inhibition showed similar protective properties but to a lesser extent. These findings may pave the way to new approaches to treat cardiovascular diseases, especially those associated with extracellular matrix degradation.

## Figures and Tables

**Figure 1 biomedicines-10-03271-f001:**
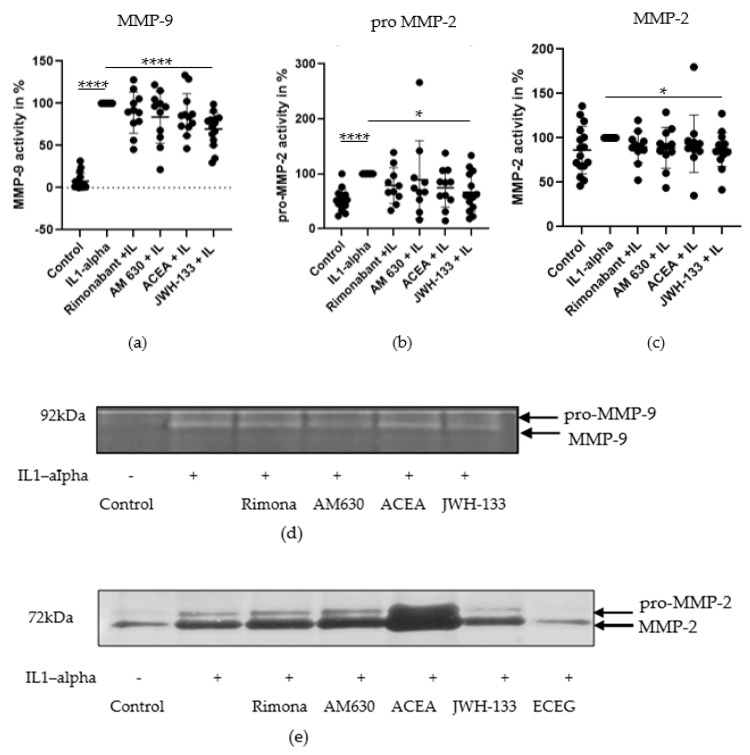
Effect of rimonabant (Rimona), AM630, ACEA and JWH-133 on IL-1α-induced secretion of MMP-9 (**a**), proMMP-2 (**b**) and MMP-2 (**c**) in VSMCs, 48 h after treatment. The *graphs* represent the *densitometric* analysis (mean ± SD; *n* = 11). Statistical analysis performed with *t*-test with Welch’s correction, * *p* < 0.05; **** *p* < 0.0001. (**d**) MMP-9 activity, gelatin zymography, VSMCs, representative zymogram. (**e**) ProMMP-2, MMP-2 activity VSMCs, representative zymogram.

**Figure 2 biomedicines-10-03271-f002:**
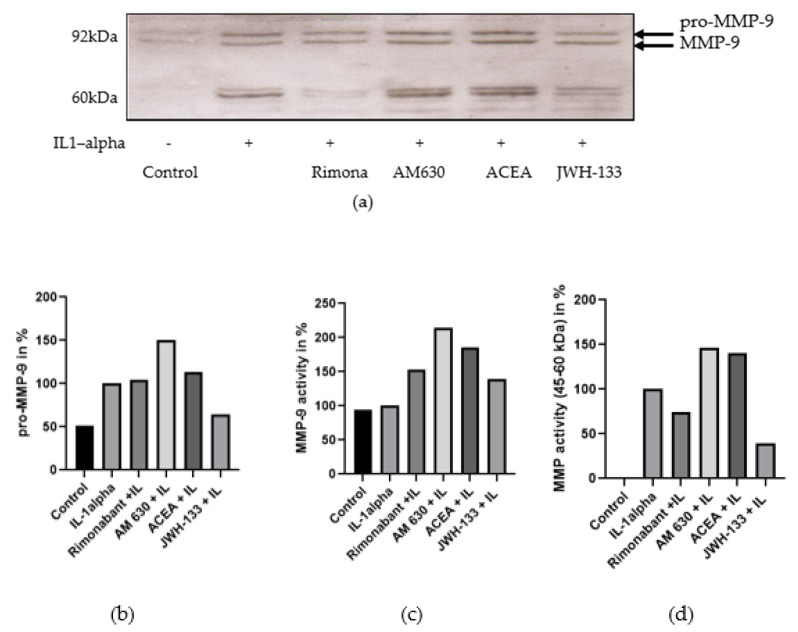
(**a**) Representative Western blots of MMP-9 in VSMCs, 48 h after stimulation with IL-1α. (**b**) Densiometric analysis of proMMP-9. (**c**) Densiometric analysis of MMP-9. (**d**) Densiometric analysis of MMP (45–60 kDa).

**Figure 3 biomedicines-10-03271-f003:**
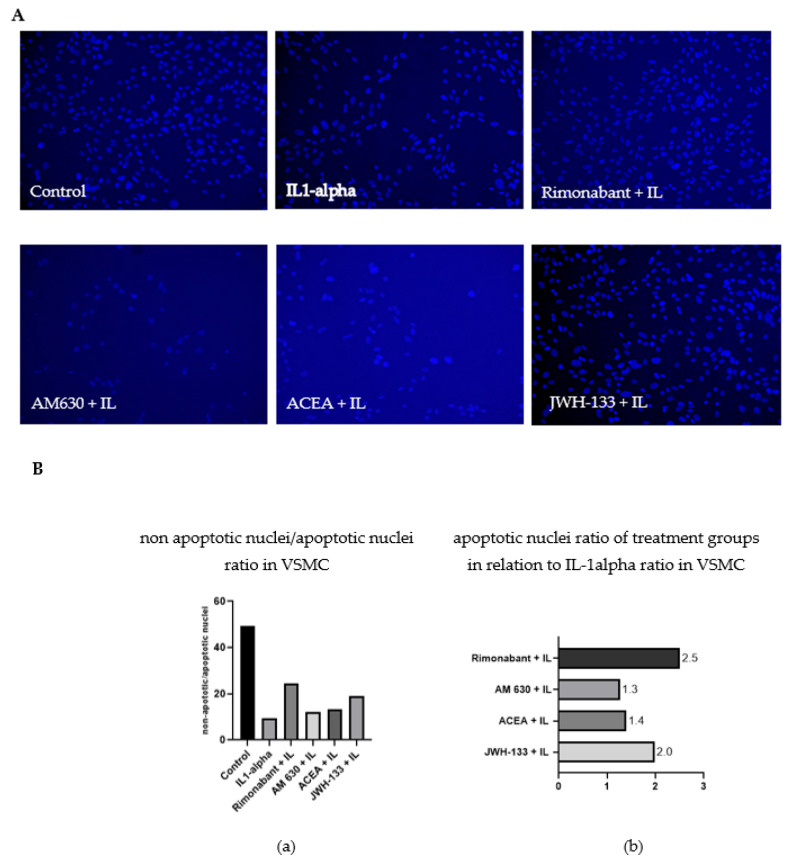
(**A**). Effect of rimonabant, AM630, ACEA and JWH-133 on IL-1α-induced apoptosis in VSMCs. Magnification ×10. More condensed small cell nuclei with irregular form (blebbing) as well as more disintegrating cells were detected after IL-1α stimulation (**B**). (**a**) The ratio of normal cell nuclei to apoptotic cell nuclei in VSMC. The higher the bar, the more normal VSMC could be found in the treatment group. (**b**) The relation of apoptotic ratio in the treatment group to the ratio of the IL-1α-stimulated group. The resulting number expresses the factor of increased normal cell nuclei in comparison to the IL-1α group.

**Figure 4 biomedicines-10-03271-f004:**
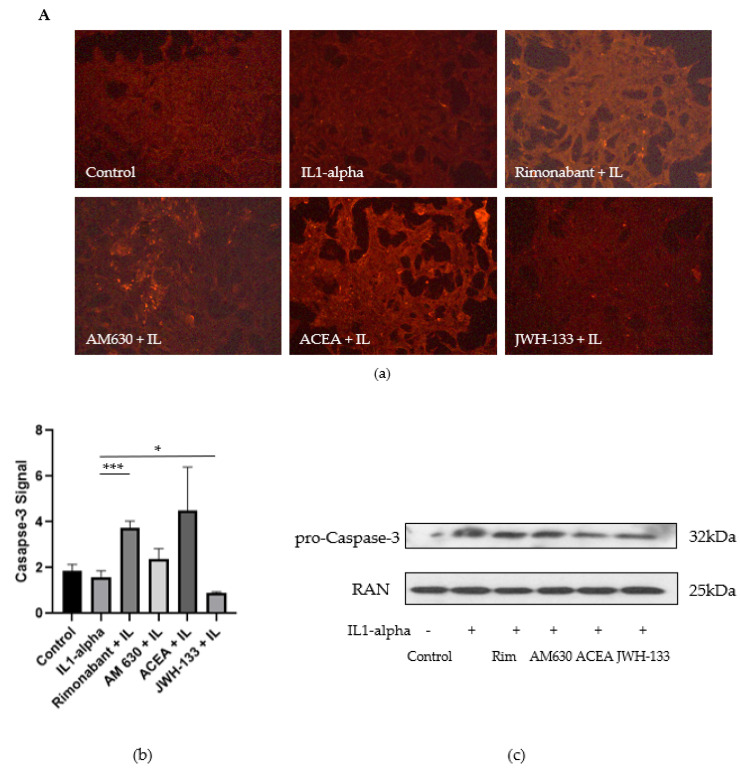
(**A**) (**a**) Caspase-3 expression signal in VSMCs, 48 h after treatment with rimonabant, AM 630, ACEA and JWH-133; fluorescent images obtained with Biorevo BZ 900 Microscope using 10× magnification. Exposure time was equal in all observed groups. (**b**) Intensity of caspase-3 expression signal. The results were attained by measuring three areas of interest of the fluorescent image and subtracting the background signal (*n* = 3). (**c**) Representative Western blots of caspase-3 in VSMCs, 48 h after stimulation with IL-1α. (**B**) (**a**) FasL expression signal in VSMCs, 48 h after treatment with rimonabant, AM 630, ACEA and JWH-133; fluorescent images obtained with Biorevo BZ 900 Microscope using 10× magnification. Exposure time was equal in all observed groups. (**b**) Intensity of FasL expression signal. The results were attained by measuring three areas of interest of the fluorescent image and subtracting the background signal (*n* = 3). (**c**) Intensity of TGF-beta1 expression signal in VSMCs, 48 h after treatment with rimonabant, AM 630, ACEA, and JWH-133, attained by measuring three areas of interest of the fluorescent image and subtracting the background signal. The measurements were repeated three times per image. Statistical testing was performed using unpaired *t*-tests. Significance was expressed when *p* < 0.05; (* *p* < 0.05; ** *p* < 0.01; *** *p* < 0.001).

**Figure 5 biomedicines-10-03271-f005:**
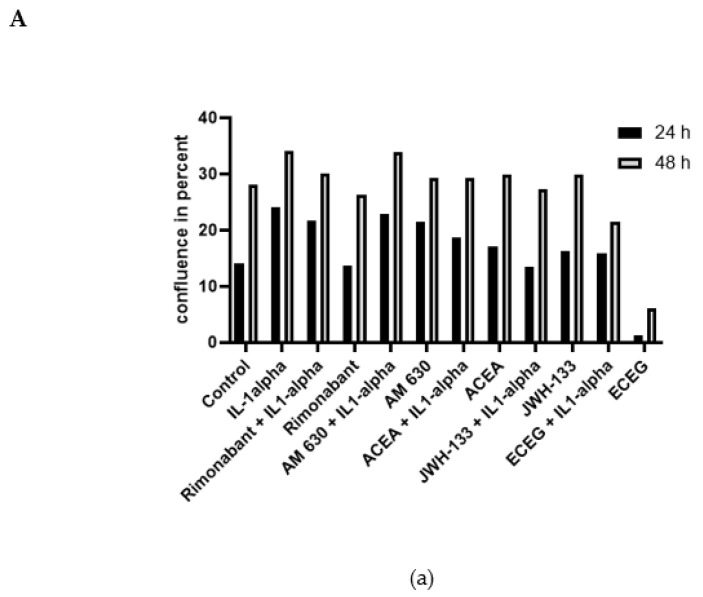
(**A**) (**a**) VSMC, cell confluence difference (cell growth) at 24 h (confluence at 24 h-confluence at 0 h) and the confluence at 48 h (confluence at 48 h-confluence at 0 h) in the treatment groups. The experiment was performed using 10% FBS medium with and without IL-1α stimulation in all treatment groups. (**b**) VSMC, cell growth in dynamic. Representative graph obtained from IncuCyte Live-Cell Analysis System. Cell proliferation was monitored by analyzing the occupied area (% confluence) of cell images over 100 h. Analysis of the IncuCyte images was performed with Incucyte^®^ Analysis Software. The experiment was performed using 10% FBS medium with and without IL-1α stimulation in all treatment groups. (**B**) The growth rate (confluence difference) of VSMCs estimated by IncuCyte live-cell analysis after treatment with compounds after IL-1α stimulation in 24 h (**a**) and 48 h (**b**). Statistical testing was performed using unpaired *t*-tests. Significance was expressed when *p* < 0.05; *n* = 6–32 (* *p* < 0.05; *** *p* < 0.001).

**Figure 6 biomedicines-10-03271-f006:**
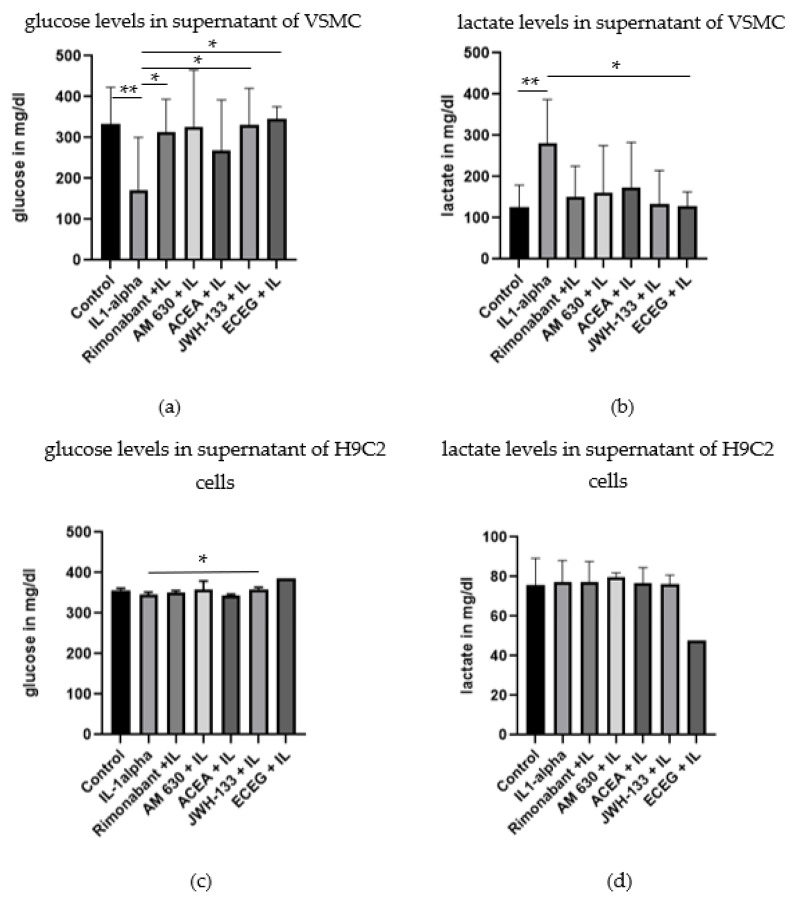
Concentration of glucose (**a**) and lactate (**b**) in the supernatant of VSMC after 48 h of treatment with compounds and IL-1α stimulation. Concentration of glucose (**c**) and lactate (**d**) in the supernatant of H9c2 cells after 48 h of treatment with compounds and IL-1α stimulation. The values are expressed as mean ± SD (VSMC, *n* = 3–9; H9C2 cells, *n* = 3). Statistical testing was performed using unpaired *t*-tests. (* *p* < 0.05; ** *p* < 0.01).

**Figure 7 biomedicines-10-03271-f007:**
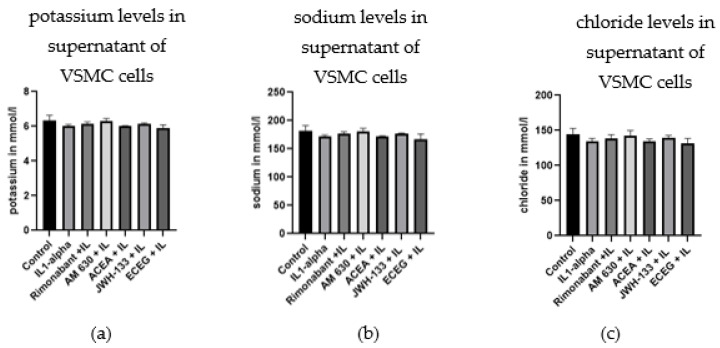
The concentrations of potassium (**a**), sodium (**b**) and chloride (**c**), measured in cell supernatant of VSMCs 48 h after treatments with compounds and IL-1α stimulation.

**Table 1 biomedicines-10-03271-t001:** Comparison of the effects of the CB1 and CB2 receptor agonists and antagonists on cytokine-induced MMPs secretion, apoptosis, glucose uptake and cell proliferation in VSMCs and cardiac H9c2 cells. ↑—increase; ↓—decrease.

Compound	IL-1α	ACEA	Rimonabant	JWH-133	AM 630	ECEG
CB receptor binding	none	CB1 agonist	CB1 antagonist	CB2 agonist	CB2 antagonist	uncertain
MMP activity VSMC	↑	no effect	↓ MMP-9 ↓ proMMP-2 by tendency	↓↓↓ MMP-9 ↓↓ proMMP-2 ↓↓ MMP-2	no effect	↓↓ MMP-9 ↓↓ proMMP-2 ↓↓ MMP-2
MMP activity H9c2	↑	↑ MMP-9	↑ MMP-9	↓ MMP-9 (13%) ↓ proMMP-2 (30%)	↑ MMP-9 (30%)	↓↓ MMP9
Apoptosis VSMC	↑	no effect	↓↓ 2.5-fold	↓↓ 2.0-fold	no effect	--------------
Apoptosis H9c2	↑	no effect	↓ 1.3-fold	↓ 1.1-fold	no effect	↓↓ 1.4-fold
Fas L VSMC	↑	↑	↓↓	↓↓	↓	---------------
Caspase-3 VSMC	↑	↑↑	no effect	↓↓	↑	---------------
TGF-beta1 VSMC	↓	no effect	↑	↑	no effect	---------------
Cell proliferation VSMC (10% FBS, 48h)	↑	no effect	no effect	no effect	no effect	↓
Glucose in cell supernatant VSMC	↓↓↓	↑(1.8-fold, n.s.)	↑ 2.1-fold vs. IL1α	↑ 2.2-fold vs. IL1α	↑(2.2-fold, n.s.)	↑ 2.4-fold vs. IL1α
Glucose in cell supernatant H9c2 cells	↓	no effect	no effect	↑ vs. IL1α	no effect	↑ n.s.

## Data Availability

The data presented in this study are available on request from the corresponding author.

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
