# Peer review of "Differential Regulation of MMPs, Apoptosis and Cell Proliferation by the Cannabinoid Receptors CB1 and CB2 in Vascular Smooth Muscle Cells and Cardiac Myocytes"

_biomedicines, 2022, doi:10.3390/biomedicines10123271_

Round 1

Reviewer 1 Report

Dear Authors

The manuscript is interesting and deserves consideration in this journal. The topic is suitable and the content is correctly organized. I suggest to performa  statystical validation of data and correlation analysis to give more importance to the data found. Also introduction could be improved in light of the existing relation between inflammation process and CB1/CB2 receptors, for example: "Exploring the first Rimonabant analog-opioid peptide hybrid compound, as bivalent ligand for CB1 and opioid receptors", "Novel Fubinaca/Rimonabant hybrids as endocannabinoid system modulators", "Preparation of bivalent agonists for targeting the mu opioid and cannabinoid receptors".

Author Response

Dear Reviewer,

We thank you for the careful evaluation of our manuscript. We performed revisions, including novel data, under close guidance of the reviewer.

  1. “I suggest to perform statistical validation of data and correlation analysis to give more importance to the data found”.

We thank the reviewer for pointing out the necessity to evaluate more precisely the data. 

An additional statistical evaluation of fluorescence data on Capase-3 and Fas L and TGF-beta1 has been performed. The results are added to the manuscript in the text (lines 343-345) and in Figure 4. Based on this analysis we could confirm the regulation of Capase-3 and Fas L and TGF-beta1 by investigated compounds. We have found a significant down-regulation of caspase-3 by Rimonabant (p=0.0009) and JWH-133 (p=0.0173) as compared with IL-1 alpha stimulated group. Cytokine induced FasL expression was also significantly decreased by Rimonabant (p=0.0013), AM-630 (p=0.0006), JWH-133 (p=0.0003) and to less extent by ACEA (p=0.0174).

A correlation analysis was performed for the experiments on cell growth rates and concentration of glucose, lactate and electrolytes. Such evaluations were possible because in these experiments we analysed the same samples. We have found that an increase in glucose levels in supernatant after the treatment with JWH-133 was correlated with the decrease of lactate (Pearson r = -0.998; p (two-tailed) = 0.0131). The correlation between glucose and lactate changes after treatment with AM 630 was also significant (r =- 0.9999; p(two-tailed) = 0.0077). No significate correlations have been found in the cell growth rates. The data were added to Methods (lines 175-177) and Results (lines 557-559).

  1. “Also introduction could be improved…..”

Thank you for this suggestion. We have added in the Introduction a description of recent findings on novel compounds that affect the CB receptors (line 48-53). The references are listed accordingly (Ref.11,12,13). 

“Currently, novel modulators of the cannabinoid system are under investigations. New compounds are able to bind to CB receptors in the low nanomolar range with a marked selectivity towards the receptors (Stefanucci, 2018). Moreover, multitargeting G-protein coupled receptors is also a promising strategy as shown for antinociception by bivalent agonists for the opioid and cannabinoid receptors (Mollica, 2017; Dvorácskó S, 2019).

Reviewer 2 Report

The article is original and very interesting, showing a high potential of future improvement of treatment of cardiac diseases. Congratulations!

I suggest to be published after few minor corrections. In References section you should follow carefully the Instructions for authors. Eg., for the greatest majority of references you have mentioned (in eng). I think is no need of such mention.

For ref 3 you should mention doi.

For ref 4 you must mention the title of the article

For ref 17, is no need to mention all authors

Author Response

Dear Reviewer, 

Thank you for the careful review and positive evaluation of our study.

We have revised the manuscript and corrected all mistakes in the Reference list. All changes are indicated in yellow.